# Tranquil Clouds: Neural Networks for Learning Temporally Coherent Features in Point Clouds

**Lukas Prantl**
Department of Computer Science
Technical University of Munich
Munich, Germany

**Nuttapong Chentanez**
NVIDIA
Bangkok, Thailand

**Stefan Jeschke**
NVIDIA
Vienna, Austria

**Nils Thuerey**
Department of Computer Science
Technical University of Munich
Munich, Germany

## Abstract

Point clouds, as a form of Lagrangian representation, allow for powerful and flexible applications in a large number of computational disciplines. We propose a novel deep-learning method to learn stable and temporally coherent feature spaces for points clouds that change over time. We identify a set of inherent problems with these approaches: without knowledge of the time dimension, the inferred solutions can exhibit strong flickering, and easy solutions to suppress this flickering can result in undesirable local minima that manifest themselves as halo structures. We propose a novel temporal loss function that takes into account higher time derivatives of the point positions, and encourages mingling, i.e., to prevent the aforementioned halos. We combine these techniques in a super-resolution method with a truncation approach to flexibly adapt the size of the generated positions. We show that our method works for large, deforming point sets from different sources to demonstrate the flexibility of our approach.

## 1 Introduction

Deep learning methods have proven themselves as powerful computational tools in many disciplines, and within it a topic of strongly growing interest is deep learning for point-based data sets. These Lagrangian representations are challenging for learning methods due to their unordered nature, but are highly useful in a variety of settings from geometry processing and 3D scanning to physical simulations, and since the seminal work of Qi Charles et al. (2017), a range of powerful inference tasks can be achieved based on point sets. Despite their success, interestingly, no works so far have taken into account time. Our world, and the objects within it, naturally move and change over time, and as such it is crucial for flexible point-based inference to take the time dimension into account. In this context, we propose a method to learn temporally stable representations for point-based data sets, and demonstrate its usefulness in the context of super-resolution.

An inherent difficulty of point-based data is their lack of ordering, which makes operations such as convolutions, which are easy to perform for Eulerian data, unexpectedly difficult. Several powerful approaches for point-based convolutions have been proposed (Qi et al., 2017; Hermosilla et al., 2018; Hua et al., 2018), and we leverage similar neural network architectures in conjunction with the permutation-invariant *Earth Mover's Distance* (EMD) to propose a first formulation of a loss for temporal coherence.

In addition, several works have recognized the importance of training point networks for localized patches, in order to avoid having the network to rely on a full view of the whole data-set for tasks that are inherently local, such as normal estimation (Qi Charles et al., 2017), and super-resolution (Yu et al., 2018a). This also makes it possible to flexibly process inputs of any size without being limited by memory requirements. Later on we will demonstrate the importance of such a patch-based approach with sets of changing cardinality in our setting. A general challenge here is to deal with varying input sizes, and for super-resolution tasks, also varying output sizes. Thus, in summary we target an extremely challenging learning problem: we are facing permutation-invariant inputs and targets of varying size, that dynamically move and deform over time. In order to enable deep learning approaches in this context, we make the following key contributions: Permutation invariant loss terms for temporally coherent point set generation; A Siamese training setup and generator architecture for point-based super-resolution with neural networks; Enabling improved output variance by allowing for dynamic adjustments of the output size; The identification of a specialized form of mode collapse for temporal point networks, together with a loss term to remove them. We demonstrate that these contributions together make it possible to infer stable solutions for dynamically moving point clouds with millions of points.

More formally, we show that our learning approach can be used for generating a point set with an increased resolution from a given set of input points. The generated points should provide an improved discretization of the underlying ground truth shape represented by the initial set of points. For the increase, we will target a factor of two to three per spatial dimension. Thus, the network has the task to estimate the underlying shape, and to generate suitable sampling positions as output.

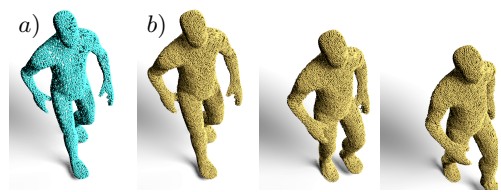

Figure 1: Our algorithm upsamples an input point cloud (a) in a temporally coherent manner. Three exemplary outputs are shown in yellow in (b).

This is generally difficult due to the lack of connectivity and ordering, and in our case, positions that move over time in combination with a changing number of input points. Hence it is crucial that the network is able to establish a temporally stable latent space representation. Although we assume that we know correspondences over time, i.e., we know which point at time $t$ moved to a new location at time $t + \Delta t$, the points can arbitrarily change their relative position and density over the course of their movement, leading to a substantially more difficult inference problem than for the static case.

## 2 RELATED WORK

Deep learning with static point sets was first targeted in PointNet (Qi Charles et al., 2017) via order-invariant networks, while PointNet++ (Qi et al., 2017) extended this concept to generate features for localized groups similar to a convolution operation for grid-based data. This concept can be hierarchically applied to the generated groups, in order to extract increasingly abstract and global features. Afterwards, the extracted features can be interpolated back to the original point cloud. The goal to define point convolutions has been explored and extended in several works. The MCNN approach (Hermosilla et al., 2018) phrased convolution in terms of a Monte Carlo integration. PointCNN (Hua et al., 2018) defined a pointwise convolution operator using nearest neighbors, while extension-restriction operators for mapping between a point cloud function and a volumetric function were used in Atzmon et al. (2018). The PU-Net (Yu et al., 2018a) proposed a network for upsampling point clouds, and proposed a similar hierarchical network structure of PointNets along the lines of PointNet++ to define convolutions. Being closer to our goals, we employ this approach for convolutional operations in our networks below. We do not employ the edge-aware variant of the PU-Net (Yu et al., 2018b) here, to keep it as simple and general as possible as we focus on temporal changes in our work.

Permutation invariance is a central topic for point data, and was likewise targeted in other works (Ravanbakhsh et al., 2016; Zaheer et al., 2017). The Deep Kd-network (Klokov and Lempitsky, 2017) defined a hierarchical convolution on point clouds via kd-trees. PointProNets (Roveri et al., 2018) employed deep learning to generate dense sets of points from sparse and noisy input points for 3D reconstruction applications. PCPNet (Guerrero et al., 2018), as another multi-scale variant of

PointNet, has demonstrated high accuracy for estimating local shape properties such as normal or curvature. P2PNet (Yin et al., 2018) used a bidirectional network and extends PointNet++ to learn a transformation between two point clouds with the same cardinality.

Recently, the area of point-based learning has seen a huge rise in interest. One focus here are 3D segmentation problems, where numerous improvements were proposed, e.g., by SPLATNet (Su et al., 2018), SGPN (Wang et al., 2018a), SpiderCNN (Xu et al., 2018), PointConv (Wu et al., 2018), SO-NEt(Li et al., 2018a) and 3DRNN (Ye et al., 2018). Other networks such as Flex Convolution (Groh et al., 2018), the SuperPoint Graph (Landrieu and Simonovsky, 2018), and the fully convolutional network (Rethage et al., 2018) focused on large scale segmentation. Additional areas of interest are shape classification (Wang et al., 2018b; Lei et al., 2018; Zhang and Rabbat, 2018; Skouson, 2018) and object detection (Simon et al., 2018; Zhou and Tuzel, 2018), and hand pose tracking (Ge et al., 2018). Other works have targeted rotation and translation invariant inference (Thomas et al., 2018), and point cloud autoencoders (Yang et al., 2018; Deng et al., 2018). A few works have also targeted generative models based on points, e.g., for point cloud generation (Sun et al., 2018), and with adversarial approaches (Li et al., 2018b). It is worth noting here that despite the huge interest, the works above do not take into account temporally changing data, which is the focus of our work. A notable exception is an approach for scene flow (Liu et al., 2018), in order to estimate 3D motion directly on the basis of point clouds. This work is largely orthogonal to ours, as it does not target generative point-based models.

## 3 METHODOLOGY

We assume an input point cloud $X = \{x_1, x_2, ..., x_k\}$ of size $k \in [1, k_{max}]$. It consists of points $x_i \in \mathbb{R}^d$, where $d$ includes 3 spatial coordinates and optionally additional features. Our goal is to let the network $f_s(X)$ infer a function $\tilde{Y}$ which approximates a desired super-resolution output point cloud $Y = \{y_1, y_2, ..., y_n\}$ of size $n \in [1, n_{max}]$ with $y_i \in \mathbb{R}^3$, i.e. $f_s(X) = \tilde{Y} \approx Y$. For now we assume that the number of output points $n$ is defined by multiplying $k$ with a user-defined upsampling factor $r$, i.e. $n = rk$. Figure 2a) illustrates the data flow in our super-resolution network schematically. We treat the upsampling problem as a *local* one, i.e., we assume that the inference problem can be solved based on a spatially constrained neighborhood. This allows us to work with individual *patches* extracted from input point clouds. At the same time, it makes it possible to upsample adaptively, for example, by limiting the inference to relevant areas, such as complex surface structures. For the patch extraction we use a fixed spatial radius and normalize point coordinates within each patch to lie in the range of $[-1, 1]$.

Our first building block is a measure for how well two point clouds represent the same object or scene by formulating an adequate spatial loss function. Following Achlioptas et al. (2017), we base our spatial loss $\mathcal{L}_S$ on the *Earth Mover's Distance* (EMD), which solves an assignment problem to obtain a differentiable bijective mapping $\phi : \tilde{y} \to y$. With $\phi$ we can minimize differences in position for arbitrary orderings of the points clouds via:

$$\mathcal{L}_S = \min_{\phi:\tilde{y}\to y} \sum_{\tilde{y}_i \in \tilde{Y}} \|\tilde{y}_i - \phi(\tilde{y}_i)\|_2^2 \qquad (1)$$

### 3.1 TEMPORAL COHERENCE

When not taking temporal coherence explicitly into account, the highly nonlinear and ill-posed nature of the super-resolution problem can cause strong variations in the output even for very subtle changes in the input. This results in significant temporal artifacts that manifest themselves as flickering. In order to stabilize the output while at the same time keeping the network structure as small and simple as possible, we propose the following training setup. Given a sequence of high resolution point clouds $Y^t$, with $t$ indicating time, we can compute a velocity $V^t = \{v_1^t, v_2^t, ..., v_k^t\}$, where $v_i^t \in \mathbb{R}^3$. For

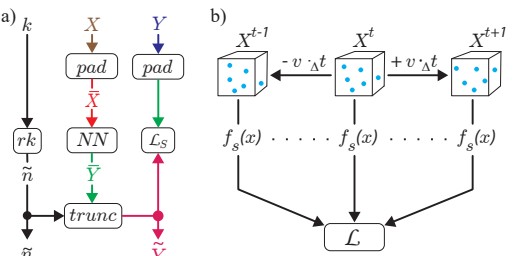

Figure 2: a) Schematic overview of $f_s(X)$. Black arrows represent scalar data. Point data is depicted as colored arrows with the color indicating data cardinality (brown=$k$, red = $k_{max}$, green = $n_{max}$, blue = $n$, and purple = $\tilde{n}$). b) Siamese network setup for temporal loss calculation.

this we use a finite difference $(y_i^{t+1} - y_i^t)$, where we assume, without loss of generality, $\Delta t = 1$, i.e. the time step is normalized to one. For training, the low resolution inputs $X$ can now be generated from $Y$ via down-sampling by a factor of $r$, which yields a subset of points with velocities. Details of our data generation process will be given below.

To train a temporally coherent network with the $Y^t$ sequences, we employ a *Siamese* setup shown in Figure 2b. We evaluate the network several times (3 times in practice) with the same set of weights, and moving inputs, in order to enforce the output to behave consistently. In this way we avoid recurrent architectures that would have to process the high resolution outputs multiple times. In addition, we can compute temporal derivatives from the input points, and use them to control the behavior of the generated output.

Under the assumption of slowly moving inputs, which theoretically could be ensured for training, a straightforward way to enforce temporal coherence would be to minimize the movement of the generated positions over consecutive time steps in terms of an $L_2$ norm:

$$\mathcal{L}_{2V} = \sum_{i=1}^{n} \|\tilde{y}_i^{t+1} - \tilde{y}_i^t\|_2^2. \tag{2}$$

While this reduces flickering, it does not constrain the change of velocities, i.e., the acceleration. This results in a high frequency jittering of the generated point positions. The jitter can be reduced by also including the previous state at time step $t - 1$ to constrain the acceleration in terms of its $L_2$ norm:

$$\mathcal{L}_{2A} = \sum_{i=1}^{n} \|\tilde{y}_i^{t+1} - 2\tilde{y}_i^t + \tilde{y}_i^{t-1}\|_2^2 \tag{3}$$

However, a central problem of a direct temporal constraint via Equations (2) and (3) is that it consistently leads to a highly undesirable clustering of generated points around the center point. This is caused by the fact that the training procedure as described so far is unbalanced, as it only encourages minimizing changes. The network cannot learn to reconstruct realistic, larger motions in this way, but rather can trivially minimize the loss by contracting all outputs to a single point. For this reason, we instead use the estimated velocity of the ground truth point cloud sequence with a forward difference in time, to provide the network with a reference. By using the EMD-based mapping $\phi$ established for the spatial loss in Equation (1), we can formulate the temporal constraint in a permutation invariant manner as

$$\mathcal{L}_{EV} = \sum_{i=1}^{n} \|\big(\tilde{y}_i^{t+1} - \tilde{y}_i^t\big) - \big(\phi(\tilde{y}_i^{t+1}) - \phi(\tilde{y}_i^t)\big)\|_2^2. \tag{4}$$

Intuitively, this means the generated outputs should mimic the motion of the closest ground truth points. As detailed for the $L_2$-based approaches above, it makes sense to also take the ground truth acceleration into account to minimize rapid changes of velocity over time. We can likewise formulate this in a permutation invariant way w.r.t. ground truth points via:

$$\mathcal{L}_{EA} = \sum_{i=1}^{n} \|\big(\tilde{y}_i^{t+1} - 2\tilde{y}_i^t + \tilde{y}_i^{t-1}\big) - \big(\phi(\tilde{y}_i^{t+1}) - 2\phi(\tilde{y}_i^t) + \phi(\tilde{y}_i^{t-1})\big)\|_2^2. \tag{5}$$

We found that a combination of $\mathcal{L}_{EV}$ and $\mathcal{L}_{EA}$ together with the spatial loss $\mathcal{L}_S$ from Eq. 1 provides the best results, as we will demonstrate below. First, we will introduce the additional loss terms of our algorithm.

## 3.2 VARIABLE POINT CLOUD SIZES

Existing network architectures are typically designed for processing a fixed amount of input and output points. However, in many cases, and especially for a localized inference of super-resolution, the number of input and output points varies significantly. While we can safely assume that no patch exceeds the maximal number of inputs $k_{max}$ (this can be ensured by working on a subset), it can easily happen that a certain spatial region has fewer points. Simply including more distant points could guarantee that we have a full set of samples, but this would mean the network has to be invariant to scaling, and to produce features at different spatial scales. Instead, we train our

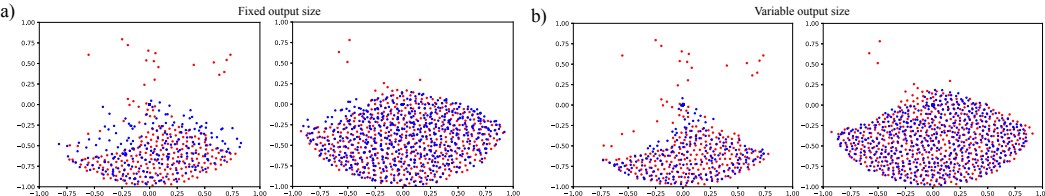

Figure 3: An illustration of the relationship between input and output size. (a,b,d) show histograms of point set sizes for: (a,b) the input set; (c) the ground truth target sets; and (d) the network output, i.e. $r$ times larger than the input. The latter deviates from the ground truth in (c), but follows its overall structure. This is confirmed in (b), which shows a heat map visualization of input vs. ground truth output size. The diagonal structure of the peak confirms the approximately linear relationship.

Figure 4: The effect of our variable output handling for exemplary patches. In red the ground truth target, in blue the inferred solution. Left (a) with fixed output size, and on the right (b) with the proposed support for variable output sizes. The latter approximates the shape of the red ground truth points significantly better. (a) leads to rather uniform shapes that, e.g., cover empty space above the ground truth in both examples.

network for a fixed spatial size, and ensure that it can process varying numbers of inputs. For inputs with fewer than $k_{max}$ points, we pad the input vector to have a fixed size. Here, we ensure that the padding values are not misinterpreted by the network as being point data. Therefore, we pad $X$ with $p \in \{-2\}^d$, which represents a value outside the regular patch coordinate range $[-1, 1]$: $\bar{X} = \{x_1, x_2, ..., x_k, \underbrace{p, p, ..., p}_{k_{max}-k}\}$. The first convolutional layer in our network now filters out the padded entries using the following mask: $M_{in} = \{m_{i \in [0,k]}\} = \{\underbrace{1, 1, ..., 1}_{k}, \underbrace{0, 0, ..., 0}_{k_{max}-k}\}$. The entries of $p$ allow us to compute the mask on the fly throughout the whole network, without having to pass through $k$. For an input of size $k$, our network has the task to generate $\tilde{n} = rk$ points. As the size of the network output is constant with $rk_{max}$, the outputs are likewise masked with $M_{out}$ to truncate it to length $\tilde{n}$ for all loss calculations, e.g., the EMD mappings. Thus, as shown in Figure 2a, $\tilde{n}$ is used to truncate the point cloud $\bar{Y} = \{\bar{y}_1, \bar{y}_2, ..., \bar{y}_{n_{max}}\}$ via a mask $M_{out}$ to form the final output $\tilde{Y} = \{\bar{y}_i | i \in [1, \tilde{n}]\}$.

Note that in Sec. 3.1, we have for simplicity assumed that $n = rk$, however, in practice the number of ground truth points $n$ varies. As such, $\tilde{n}$ only provides an approximation of the true number of target points in the ground truth data. While the approximation is accurate for planar surfaces and volumes, it is less accurate in the presence of detailed surface structures that are smaller than the spatial frequency of the low-resolution data.

We have analyzed the effect of this approximation in Fig. 3. The histograms show that the strongly varying output counts are an important factor in practice, and Fig. 4 additionally shows the improvement in terms of target shape that results from incorporating variable output sizes. In general, $\tilde{n}$ provides a good approximation for our data sets. However, as there is a chance to infer an improved estimate of the correct output size based on the input points, we have experimented with training a second network to predict $\tilde{n}$ in conjunction with a differentiable output masking. While this could be an interesting feature for future applications, we have not found it to significantly improve results. As such, the evaluations and results below will use the analytic calculation, i.e., $\tilde{n} = rk$.

## 3.3 PREVENTING HALO ARTIFACTS

For each input point the network generates $r$ output points, which can be seen as individual groups $g$: $\psi(g) = \{\tilde{Y}_i | i \in [rg + 1, (r + 1)g]\}$. These groups of size $r$ in the output are strongly related to the input points they originate from. Networks that focus on maintaining temporal coherence for the dynamically changing output tend to slide into local minima where $r$ output points are attached as a

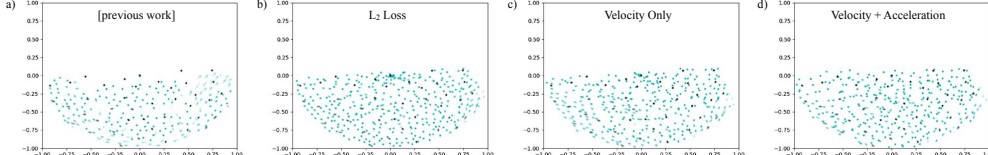

Figure 5: Ablation study for our temporal loss formulation. Black points indicate targets, while green points are generated (both shown as time average). a) Result from previous work; b) With $L_{2V}$ loss; c) the proposed velocity loss $L_{EV}$; d) our full loss formulation with $L_{EV} + L_{EA}$. While (a) has difficulties approximating the target shape and the flickering output is visible as blurred positions, the additional loss terms (esp. in (c) and (d)) provide stable results that closely approximate the targets. Note that (b) leads to an undesirably static motion near the bottom of the patch. As the input points here are moving the output should mimic this motion, like (c,d).

fixed structure to the input point location. This manifests itself as visible static halo-like structures that move along with the input. Although temporal coherence is good in this case, these cluster-like structures lead to gaps and suboptimal point distributions in the output, particularly over time. These structures can be seen as a form of temporal mode collapse that can be observed in other areas of deep learning, such as GANs. To counteract this effect, we introduce an additional *mingling* loss term to prevent the formation of clusters by pushing the individual points of a group apart:

$$\mathcal{L}_M = \frac{1}{\lceil \frac{\tilde{n}}{r} \rceil} \sum_{i}^{\lceil \frac{\tilde{n}}{r} \rceil} \frac{|\psi(i)|}{\sum_{\tilde{y}_g \in \psi(i)} \| \frac{\sum \psi(i)}{|\psi(i)|} - \tilde{y}_g \|_2} \tag{6}$$

Note that in contrast to previously used repulsion losses (Yu et al., 2018a), $\mathcal{L}_M$ encourages points to globally mix rather than just locally repelling each other. While a repulsion term can lead to a deterioration of the generated outputs, our formulation preserves spatial structure and temporal coherence while leading to well distributed points, as is illustrated in Fig. 6.

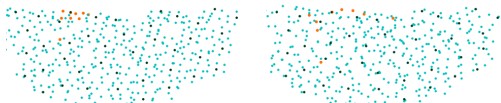

Figure 6: Left, a result without the mingling loss from Eq. 6, right with (a single point group highlighted in orange). The former has many repeated copies of a single pattern, which the mingling loss manages to distribute as can be seen in the right picture.

In combination with the spatial and temporal terms from above, this leads to our final loss function $\mathcal{L}_{final} = \mathcal{L}_S + \gamma \mathcal{L}_{EV} + \mu \mathcal{L}_{EA} + \nu \mathcal{L}_M$, with weighting terms $\gamma, \mu, \nu$.

## 4 EVALUATION AND RESULTS

We train our network in a fully supervised manner with simulated data. To illustrate the effect of our temporal loss functions, we employ it in conjunction with established network architectures from previous work (Qi Charles et al., 2017; Yu et al., 2018a). Details of the data generation and network architectures are given in the appendix. We first discuss our data generation and training setup, then illustrate the effects of the different terms of our loss function, before showing results for more complex 3D data sets. As our results focus on temporal coherence, which is best seen in motion, we refer readers to the supplemental materials at https://ge.in.tum.de/publications/2020-iclr-prantl/ in order to fully evaluate the resulting quality.

**Ablation Study** We evaluate the effectiveness of our loss formulation with a two dimensional ablation study. An exemplary patch of this study is shown in Fig. 5. In order to compare our method to previous work, we have trained a previously proposed method for point-cloud super-resolution, the PU-Net (Yu et al., 2018a) which internally uses a PointNet++ (Qi et al., 2017), with our data set, the only difference being that we use zero-padding here. This architecture will be used in the following comparisons with previous work. Fig. 5a) shows a result generated with this network. As this figure contains an average of multiple frames to indicate temporal stability, the blurred regions, esp. visible on the right side of Fig. 5a), indicate erroneous motions in the output. For this network the difficulties of temporally changing data and varying output sizes additionally lead to a suboptimal approximation

|  | $\mathcal{L}_S$ | $\mathcal{L}_N$ | $\mathcal{L}_M$ | $\mathcal{L}_{2V}$ | $\mathcal{L}_{2A}$ | $\mathcal{L}_{EV}$ | $\mathcal{L}_{EA}$ |
|---|---|---|---|---|---|---|---|
| **2D Previous work** | 0.0784 | 0.329 | 5.499 | 0.1 | 0.402 | 0.107 | 0.214 |
| **2D With $\mathcal{L}_{2V}$** | 0.044 | 0.00114 | 2.197 | 1.1e-05 | 4.2e-05 | 0.00197 | 0.00276 |
| **2D Only $\mathcal{L}_{EV}$** | 0.0453 | 0.00114 | 2.713 | 2.6e-05 | 6.0e-06 | 6.15e-04 | 5.27e-04 |
| **2D Full** | 0.0487 | 0.00116 | 3.0307 | 2.1e-05 | 1.0e-06 | 6.52e-04 | 1.46e-04 |
| **3D Previous work** | 0.0948 | 0.494 | 10.558 | 0.325 | 1.299 | 0.19 | 0.365 |
| **3D Full** | 0.0346 | 0.00406 | 3.848 | 8.04e-04 | 2.0e-06 | 0.00179 | 7.09e-04 |

Table 1: Quantitative results for the different terms of our loss functions, first for our 2D ablation study and then for our 3D versions. The first three columns contain spatial, the next four temporal metrics. $\mathcal{L}_N = \|\tilde{n} - n\|_2^2$ is given as a measure of accuracy in terms of the size of the generated outputs (it is not part of the training).

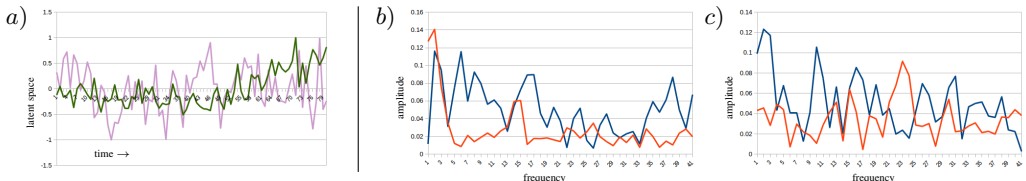

Figure 7: Illustrations of the latent spaces learned by our networks. (a) shows averaged latent space values for 100 random patch sequences of our 2D data set. The green curve shows our method with temporal coherence loss, while the pink curve was generated without it. The same data is shown in frequency space in (b), where the red curve represents the frequency of the data with temporal loss, and the blue curve the frequency of the data without. This graph highlights the reduced amount of high frequency changes in the latent space with temporal loss, esp. in frequency space, where the red curve almost entirely lies below the blue one. (c) contains frequency information for the latent space content of the same 100 patch sequences, but in a random order. In this case, the blue and red curve both contain significant amounts of high-frequencies. I.e., our method reliably identifies strongly changing inputs.

of the target points, that is also visible in terms of an increased $\mathcal{L}_S$ loss in Table 1. While Fig. 5b) significantly reduces motions, and leads to an improved shape as well as $\mathcal{L}_S$ loss, its motions are overly constrained. E.g., at the bottom of the shown patch, the generated points should follow the black input points, but in (b) the generated points stay in place. In addition, the lack of permutation invariance leads to an undesirable clustering of generated points in the patch center. Both problems are removed with $\mathcal{L}_{EV}$ in Fig. 5c), which still contains small scale jittering motions, unfortunately. These are removed by $\mathcal{L}_{EA}$ in Fig. 5d), which shows the result of our full algorithm. The success of our approach for dynamic output sizes is also shown in the $\mathcal{L}_N$ column of Table 1, which contains an $L_2$ error w.r.t. ground truth size of the outputs.

**Temporally Coherent Features**   A central goal of our work is to enable the learning of features that remain stable over time. To shed light on how our approach influences the established latent space, we analyze its content for different inputs. The latent space in our case consists of a 256-dimensional vector that contains the features extracted by the first set of layers of our network. Fig. 7 contains a qualitative example for 100 randomly selected patch sequences from our test data set, where we collect input data by following the trajectory of each patch center for 50 time steps to extract coherent data sets. Fig. 7a) shows the averaged latent space content over time for these sequences. While the model trained with temporal coherence (green curve) is also visually smoother, the difference becomes clearer when considering temporal frequencies. We measure averaged frequencies of the latent space dimensions over time, as shown in Fig. 7b,c). We quantify the differences by calculating the integral of the frequency spectrum $\tilde{f}$, weighted by the frequency $x$ to emphasize high frequencies, i.e, $\int_x x \cdot \tilde{f}(x)dx$. Hence, small values are preferred. As shown in Fig. 7b), the version trained without our loss formulations contains significantly more high frequency content. This is also reflected in the weighted integrals, which are 36.56 for the method without temporal loss, and 16.98 for the method with temporal loss. To verify that our temporal model actually establishes a stable temporal latent space instead of ignoring temporal information altogether, we evaluate the temporal frequencies for the same 100 inputs as above, but with a randomized order over time. In this case, our model correctly identifies the incoherent inputs, and yields similarly high frequencies as the regular model with 28.44 and 35.24, respectively. More details in Appendix C.

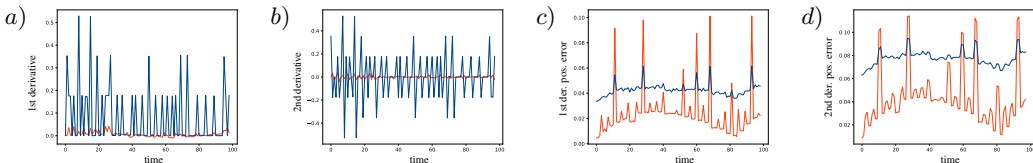

Figure 8: Evaluation of the temporal stability for generated point clouds, in red with our temporal loss formulation, in blue without. Graph (a) shows the temporal change of the point density (1st derivative), while (b) shows the 2nd derivative. In (c) and (d) the error of the 1st and 2nd derivatives of the positions w.r.t. ground-truth reference points is shown.

In addition, we evaluated the changes of generated outputs over time w.r.t. ground truth motion. For this we mapped the generated point clouds $\tilde{Y}^t = \{\tilde{y}_1^t, \tilde{y}_2^t, ..., \tilde{y}_{\tilde{n}}^t\}$ for 100 frames to evenly and dense sampled ground-truth points on the original mesh $Y^t = \{y_1^t, y_2^t, ..., y_n^t\}$ (the moving man shown in Fig. 1). This gives us a dense correlation between the data and the generated point clouds. For the mapping we used an assignment based on nearest neighbors: $\gamma : \tilde{y} \to y$. Using $\gamma$ we divide $\tilde{Y}^t$ into $n$ subsets $\hat{Y}_i = \{\tilde{y}_j | \gamma(\tilde{y}_j) = y_i\}$ which correlate with the corresponding ground-truth points. For each subset we can now compute the mean position $\frac{1}{|\hat{Y}_i|} \sum_{\hat{y} \in \hat{Y}_i} \hat{y}$ and the sample density $|\hat{Y}_i|$ measured by the number of points assigned to a ground-truth sample position. The temporal change of these values are of particular interest. The change of the mean positions should correspond to the ground-truth changes, while the change of the density should be one. We have evaluated the error of the first and second derivative of positions, as well as the first and second derivative of density (see Fig. 8 and Table 2). As can be seen from the plots, our method leads to clear improvements for all measured quantities. The individual spikes that are visible for both versions in the position errors (c,d) most likely correspond to sudden changes of the input motions for which our networks undershoots by producing a smooth version of the motion.

|  | w/o | with |  |  | w/o | with |
|---|---|---|---|---|---|---|
| **Velocity** | 0.043 | 0.024 | **Variance of 1st Derivative** | | 0.016 | 0.00013 |
| **Acceleration** | 0.078 | 0.043 | **Variance of 2nd Derivative** | | 0.038 | 0.00017 |

Table 2: Measurements averaged over 100 frames for a version of our network without temporal loss ("w/o") and with our full temporal loss formulation ("with"). The left table shows the results for the error evaluation of the velocity and the acceleration, whereas in the right table one can see the variance of the density derivatives.

**3D Results** Our patch-based approach currently relies on a decomposition of the input volumes into patches over time, as outlined in Appendix A. As all of the following results involve temporal data, full sequences are provided in the accompanying video. We apply our method to several complex 3D models to illustrate its performance. Fig. 9 shows the input as well as several frames generated with our method for an animation of a spider. Our method produces an even and temporally stable reconstruction of the object. In comparison, Fig. 9b) shows the output from the previous work architecture (Yu et al., 2018a). It exhibits uneven point distributions and outliers, e.g., above the legs of the spider, in addition to uneven motions.

A second example for a moving human figure is shown in Fig. 1. In both examples, our network covers the target shape much more evenly despite using much fewer points, as shown in Table 3. Thanks to the flexible output size of our network, it can adapt to sparsely covered regions by generating correspondingly fewer outputs. The previous work architecture, with its fixed output size, needs to concentrate the fixed number of output points within the target shape, leading to an unnecessarily large point count. In order to demonstrate the flexibility of our method, we also apply it to a volumetric moving point cloud obtained from a liquid simulation. Thanks to the patch-based evaluation of our network, it is agnostic to the overall size of the input volume. In this way, it can be used to generate coherent sets with millions of points. These examples also highlight our method's capabilities for generalization. While the 3D model was only trained on data from physics simulations, as outlined above, it learns stable features that can be flexibly applied to volumetric as well as to surface-based data. The metrics in Table 1 show that for both 2D and 3D cases, our method leads to significantly improved quality, visible in lower loss values for spatial as well as temporal terms.

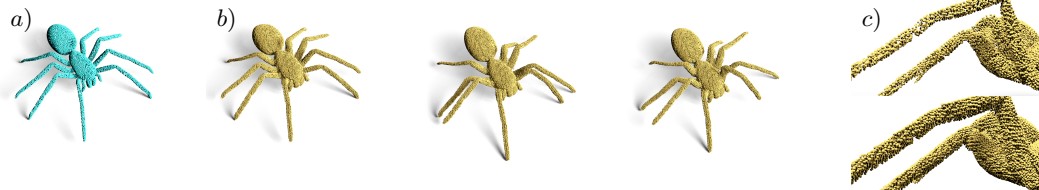

Figure 9: Our method applied to an animation of a moving spider. (a) Input point cloud, (b) three frames of our method, (c) a detail from previous work (top) and our method (bottom). Note that our method at the bottom preserves the shape with fewer outliers, and leads to a more even distribution of points, despite generating fewer points in total (see Table 3).

|  | Input points | P.W., output points | P.W., factor | Ours, output points | Ours, factor |
|---|---|---|---|---|---|
| **Spider** | 7,900 | 3,063,704 | 387.81 | 251,146 | 31.79 |
| **Moving person** | 10,243 | 5,224,536 | 510.06 | 367,385 | 35.87 |
| **Liquid** | 513,247 | - | - | 6,430,984 | 12.53 |

Table 3: Point counts for the 3D examples of our video. Input counts together with output counts for previous work (P.W.) and our proposed network are shown. Factor columns contain increase in point set size from in- to output. As previous work cannot handle flexible output counts, a fixed number of points is generated per patch, leading to a huge number of redundant points. However, our network flexibly adapts the output size and leads to a significantly smaller number of generated points that cover the object or volume more evenly.

Another interesting field of application for our algorithm are physical simulations. Complex simulations such as fluids, often employ particle-based representations. On the one hand, the volume data is much larger than surface-based data, which additionally motivates our dynamic output. On the other hand, time stability plays a very important role for physical phenomena. Our method produces detailed outputs for liquids, as can be seen in our supplemental video.

Convergence graphs for the different versions are shown in Fig. 12 of the supplemental material. These graphs show that our method not only successfully leads to very low errors in terms of temporal coherence, but also improves spatial accuracy. The final values of $\mathcal{L}_S$ for the 2D case are below 0.05 for our algorithm, compared to almost 0.08 for previous work. For 3D, our approach yields 0.04 on average, in contrast to ca. 0.1 for previous work.

## 5  CONCLUSION

We have proposed a first method to infer temporally coherent features for point clouds. This is made possible by a novel loss function for temporal coherence in combination with enabling flexible truncation of the results. In addition we have shown that it is crucial to prevent static patterns as easy-to-reach local minima for the network, which we avoid with the proposed a mingling loss term. Our super-resolution results above demonstrate that our approach takes an important first step towards flexible deep learning methods for dynamic point clouds.

Looking ahead, our method could also be flexibly combined with other network architectures or could be adopted for other applications. Specifically, a combination with PSGN (Fan et al., 2016) could be used to generate point clouds from image sequences instead of single images. Other conceivable applications could employ methods like Dahnert et al. (2019) with our approach for generating animated meshes. Due to the growing popularity and ubiquity of scanning devices it will, e.g., be interesting to investigate classification tasks of 3D scans over time as future work. Apart from that, physical phenomena such as elastic bodies and fluids (Li et al., 2019) can likewise be represented in a Lagrangian manner, and pose interesting challenges and complex spatio-temporal changes.

ACKNOWLEDGMENTS

This work is supported by grant TH 2034/1-1 of the Deutsche Forschungsgemeinschaft (DFG).

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

# Tranquil Clouds: Neural Networks for Learning Temporally Coherent Features in Point Clouds, *Supplemental Material*

## A  TRAINING AND EVALUATION MODALITIES

**Data Generation**   We employ a physical simulation to generate our input and output pairs for training. This has the advantage that it leads to a large variety of complex motions, and gives full control of the generation process. More specifically, we employ the IISPH (Ihmsen et al., 2014) algorithm, a form of Lagrangian fluid simulator that efficiently generates incompressible liquid volumes. These simulations also have the advantage that they inherently control the density of the point sampling thanks to their volume conserving properties. In order to generate input pairs for training, we randomly sample regions near the surface and extract points with a given radius around a central point. This represents the high-resolution target. To compute the low-resolution input, we downsample the points with a Poisson-disk sampling to compute a point set with the desired larger spacing. In order to prevent aliasing from features below the coarse resolution, we perform a pass of surface fairing and smoothing before downsampling. Due to the large number of patches that can be extracted from these simulations, we did not find it necessary to additionally augment the generated data sets. Examples of the low- and high-resolution pairs are shown in the supplemental material.

Below we will demonstrate that models trained with this data can be flexibly applied to moving surface data as well as new liquid configurations. The surface data is generated from animated triangle meshes that were resampled with bicubic interpolation in order to match a chosen average per-point area. This pattern was generated once and then propagated over time with the animation. When applying our method to new liquid simulations, we do not perform any downsampling, but rather use all points of a low-resolution simulation directly, as a volumetric re-sampling over time is typically error prone, and gives incoherent low resolution inputs.

Given a moving point cloud, we decompose it into temporally coherent patches in the following manner: We start by sampling points via a Poisson-disk sampling in a narrow band around the surface, e.g., based on a signed distance function computed from the input cloud. These points will persist as patch centers over time, unless they move too close to others, or too far away from the surface, which triggers their deletion. In addition, we perform several iterations for every new frame to sample new patches for points in the input cloud that are outside all existing patches. Note that this resampling of patches over time happens instantaneously in our implementation. While a temporal fading could easily be added, we have opted for employing transitions without fading, in order to show as much of the patch content as possible.

**Network Architecture and Training**   Our architecture heavily relies on a form of hierarchical point-based convolutions. I.e., the network extracts features for a subset of the points and their nearest neighbors. For the point convolution, we first select a given number of group centers that are evenly distributed points from a given input cloud. For each group center, we then search for a certain number of points within a chosen radius (a fraction of the [-1,1] range). This motivates our choice for a coordinate far outside the regular range for the padded points from Sec. 3.2. They are too far away from all groups by construction, so they are filtered out without any additional overhead. In this way, both feature extraction and grouping operations work flexibly with the varying input sizes. Each group is then processed by a PointNet-like sub-structure (Qi Charles et al., 2017), yielding one feature vector per group.

The result is a set of feature vectors and the associated group position, which can be interpreted as a new point cloud to repeatedly apply a point convolution. In this way, the network extracts increasingly abstract and global features. The last set of features is then interpolated back to the original points of the input. Afterwards a sub-pixel convolution layer is used to scale up the point cloud extended with features and finally the final position vectors are generated with the help of two additional shared, fully-connected layers. While we keep the core network architecture unmodified to allow for comparisons with previous work, an important distinction of our approach is the input and output masking, as described in Sec. 3.2.

Our point data was generated with a mean point spacing, i.e., Poisson disk radius, of $0.5$ units. For the 2D tests, an upscaling factor of $r = 9$ was used. For this purpose, patches with a diameter of 5 were extracted from the low-resolution data and patches with a diameter of 15 from the high-resolution

data. We used the thresholds $k_{max} = 100$ and $n_{max} = 900$. For the loss, we used $\gamma = 10$, $\mu = 10$, and $\nu = 0.001$. The network was trained with 5 epochs for a data set with 185k pairs, and a batch size of 16, the learning rate was 0.001 with a decay of 0.003. For the 3D results below, the scaling factor $r$ was set to 8. The diameter of the patches was 6 for the low-resolution data and 12 for the high-resolution data, with $k_{max} = 1280$ and $n_{max} = 10240$. The loss parameters were $\gamma = \mu = 5$, with $\nu = 0.001$. Learning rate and decay were the same for training, but instead we used 10 epochs with 54k patches in 3D, and a batch size of 4.

## B  NETWORK ARCHITECTURE DETAILS

The input feature vector is processed in the first part of our network, which consists of four point convolutions. We use $(n_g, r_g, [l_1, ..., l_d])$ to represent a level with $n_g$ groups of radius $r_g$ and $[l_1, ..., l_d]$ the $d$ fully-connected layers with the width $l_i (i = 1, ..., d)$. The parameters we use are $(k_{max}, 0.25, [32, 32, 64])$, $(k_{max}/2, 0.5, [64, 64, 128])$, $(k_{max}/4, 0.6, [128, 128, 256])$ and $(k_{max}/8, 0.7, [256, 256, 512])$. We then use interpolation layers to distribute the features of each convolution level among the input points. In this step, we reduce the output of each convolution layer with one shared, fully-connected layer per level, to a size of 64 and then distribute the features to all points of the input point cloud depending on their position. This extends the points of our original point cloud with 256 features. Fig. 11 shows a visual overview of the data flow in our network.

Afterwards, we process the data in $r$ separate branches consisting of two shared, fully interconnected layers with 256 and 128 nodes. The output is then processed with two shared fully-connected layers of 64 and 3 nodes. Finally, we add our resulting data to the input positions that have been repeated $r$ times. This provides an additional skip connection which leads to slightly more stable results. All convolution layers and fully interconnected layers use a tanh() activation function.

For the input feature vector, we make use of additional data fields in conjunction with the point positions. Our network also accepts additional features such as velocity, density and pressure of the SPH simulations used for data generation. For inputs from other sources, those values could be easily computed with suitable SPH interpolation kernels. In practice, we use position, velocity and pressure fields. Whereas the first two are important (as mentioned in Sec. 3.1), the pressure fields turned out to have negligible influence.

## C  FREQUENCY EVALUATION OF LATENT SPACE

In this section we give details for the frequency evaluation of Sec. 4. In order to measure the stability of the latent space against temporal changes, we evaluated the latent space of our network with and without temporal loss, once for 100 ordered patch sequences and once for 100 un-ordered ones. The central latent space of our network consists of the features generated by the point-convolution layers in the first part of the network and is 256 dimensional (see Fig. 11). To obtain information about its general behavior, we average the latent space components over all 100 patch sequences, subtract the mean, and normalize the resulting vector w.r.t. maximum value for each data set. The result is a time sequence of scalar values representing the mean deviations of the latent space. The Fourier transform of these vectors $\tilde{f}$, are shown in Fig. 7, and were used to compute the weighted frequency content $\int_x x \cdot \tilde{f}(x) dx$. Here, large values indicate strong temporal changes of the latent space dimensions. The resulting values are given in the main document, and highlight the stability of the latent space learned by our method.

## D  TRAINING DATA AND GRAPHS

Two examples with ground truth points and down-sampled input versions are shown in Fig. 10.

Additionally, Fig. 12 shows loss graphs for the different versions shown in the main text: 2D previous work, our full algorithm in 2D, as well as both cases for 3D. The mingling loss $\mathcal{L}_M$ is only shown as reference for the previous work versions, but indicates the strong halo-like patterns forming for the architectures based on previous work.

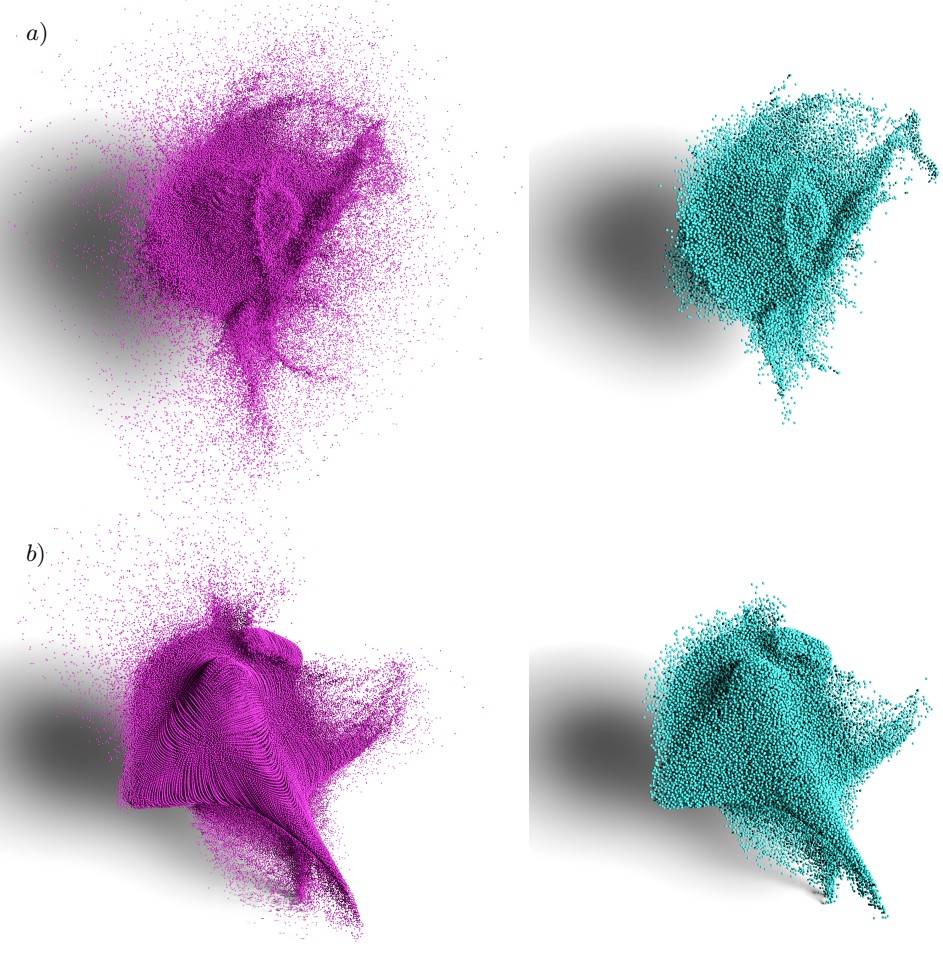

Figure 10: Examples from our synthetic data generation process. In both sections (a) and (b) a high resolution reference frame is shown in purple, and in green the down-sampled low resolution frames generated from it. The training data is generated by sampled patches from these volumes.

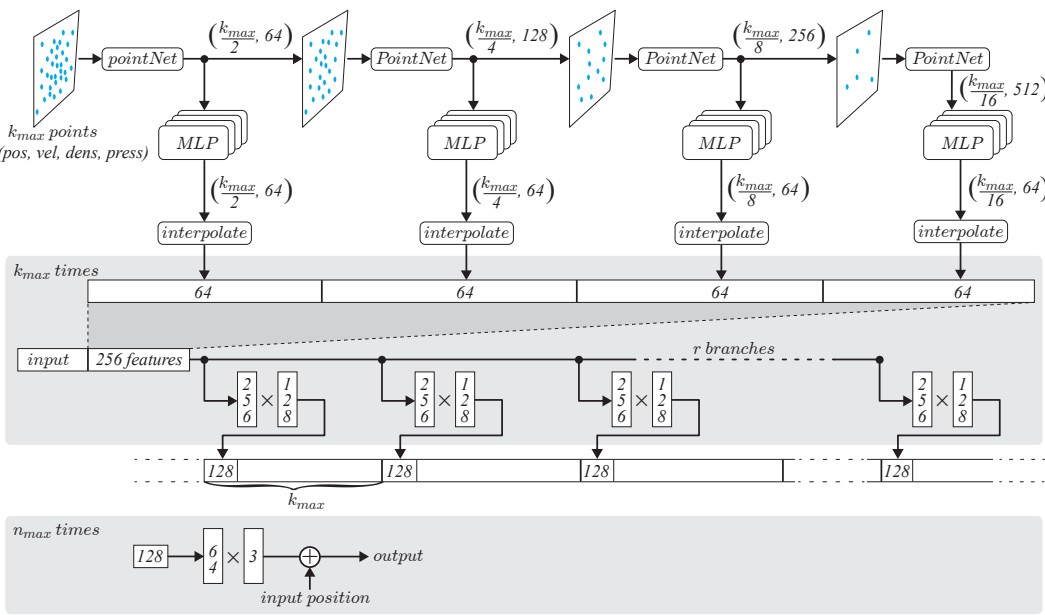

Figure 11: An overview of our network architecture. The first row shows the hierarchical point convolutions, while the bottom rows illustrate the processing of extracted features until the final output point coordinates are generated.

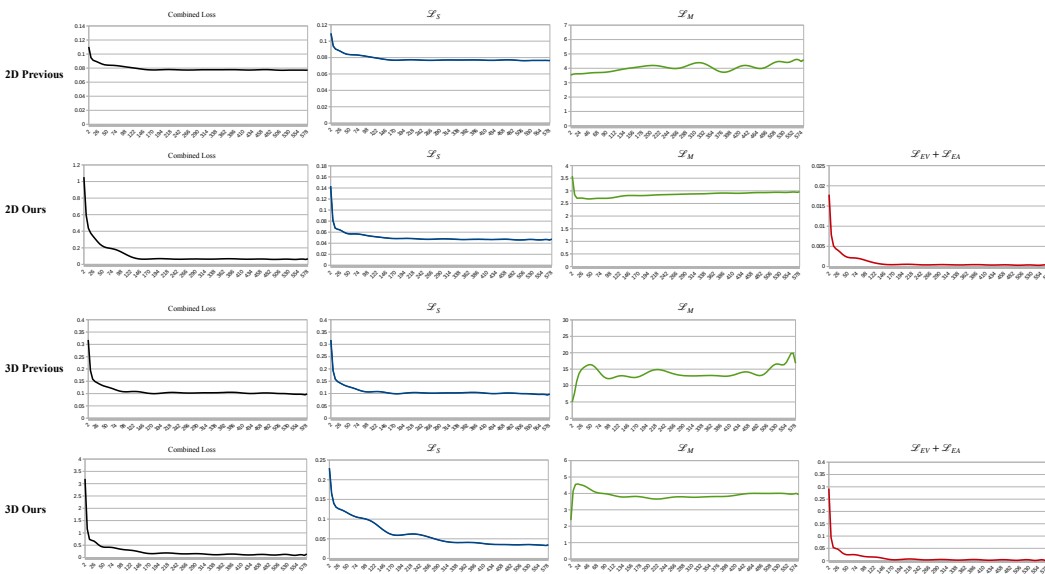

Figure 12: Convergence plots for the training runs of our different 2D and 3D versions. The combined loss only illustrates convergence behavior for each method separately, as weights and terms differ across the four variants. $\mathcal{L}_M$ for previous work is not minimized, and only given for reference.

