# OpenReview forum: "Tranquil Clouds: Neural Networks for Learning Temporally Coherent Features in Point Clouds"
_ICLR.cc/2020/Conference — Accept (Spotlight)_

### Official Review · AnonReviewer2 · 2019-10-23
**Official Blind Review #2**

**Rating:** 6

**Review:**

Summary:
This paper proposed a deep network for point cloud sequence super-resolution/upsampling. Building on the basis of pointNet and PU-net, the main contribution of the paper is identifying the problem of temporal incoherence in the process of upsampling a point cloud shape representation as well as a training loss to encourage temporal coherence. In the cases showed in the paper, the proposed method seems effective comparing to previous work which is not done on sequence data. My main concern about the work is that the experimental evaluation is limited.

Strengths:
Interesting problem and novel idea.
The proposed method is technically sound. From the provided results, the newly introduced training loss seems effective: the result sequences are visually more plausible and smooth.
Weaknesses:
Qualitative results are limited and in most cases seemingly simple. In the paper as well as the companion video, there are very few examples provided. The scale of the evaluation demonstration is not convincing enough for the readers that this work could be generalized to more complicated testing scenarios.
Quantitative results are also limited. Since the method is handling the coherence of shape deformation over time, it would be much more convincing and helpful to introduce a dense-correspondence evaluation as a benchmark. For example, one can create ground-truth correspondence from parametric morphable models and evaluate the coherence of the sequence by comparing the generated results with the ground truth.


**Experience Assessment:**

I do not know much about this area.

**Review Assessment: Checking Correctness Of Derivations And Theory:**

I assessed the sensibility of the derivations and theory.

**Review Assessment: Checking Correctness Of Experiments:**

I assessed the sensibility of the experiments.

**Review Assessment: Thoroughness In Paper Reading:**

I read the paper at least twice and used my best judgement in assessing the paper.

---

> ### Author Response · Authors · 2019-11-13
> **Thanks for your review.**
>
> We thank you for the positive assessment and feedback. Below are our responses to the concerns mentioned in your review. We will upload a revised version of our submission that extends the evaluation with respect to inputs with dense correspondences.
>
> Regarding the qualitative results we provide:
>
> - Each of our three scenarios contains between 60 and 200 frames. In total we show outputs for 420 frames with up to half a million particles. Thus the animations contain a much larger range of input configurations than most previous works. They also contain a wide range of temporal configurations that highlight the stability of our generative model. We have tested our method in other settings, and the performance of these 400+ frames is representative. Furthermore, the network used for all 3D tests was trained only once, which illustrates how well our network generalizes.
>
> It is also worth noting that due to the patch-based nature of our approach, the input data correlates only partially. While the changes look simple to our eyes, the network has to deal with many non-trivial temporal changes in the inputs (the isolated patch examples in our videos illustrate this). If the reviewers agree that additional examples are would strengthen the submission, we’d be happy to apply our network to additional input sequences.
>
> Quantitative results in our submission:
>
> - Thank you for this interesting suggestion. We performed an evaluation based on your comments: we use an animated mesh as a basis to compute ground truth reference points over time, and project the generated point positions onto the mesh. In this way we can establish a correlation between the generated point clouds, and calculate how much the change in position of the points corresponds to the ground-truth velocity and acceleration. Additionally, we also evaluate the point density, which is a good indicator of temporal stability, as it should be as uniform as possible.
>
> Velocity and acceleration are computed via the 1st and 2nd derivative of the predicted positions to the ground-truth position. These derivatives are especially important to highlight discontinuous motions and other temporal instabilities. For the density evaluation we also consider the variance of the 1st and 2nd derivative of the particle density. The variance highlights especially well how much the particle distributions vary over time.
>
> We list the resulting measurements averaged over 100 frames in the following table for a version of our network without temporal loss (“w/o”) and with our full temporal loss formulation (“with”):
>
>                             w/o          with
> Velocity            |   0.043   |   0.024
> Acceleration    |   0.078   |   0.043
>
> And for the variances of density derivatives:
>
>                                     w/o            with
> Var. of 1st Deriv.  |  0.01600  |   0.00013
> Var. of 2nd Deriv. |  0.03800  |   0.00017
>
> Thus, in both cases, our temporal loss formulation leads to substantial improvements in terms of accuracy and stability. This evaluation shows how well our algorithm approximates the ground-truth velocity, and that it generates very uniform and stable point clouds. We will include these results in our revised paper (as Figure 9).

---

### Official Review · AnonReviewer3 · 2019-10-27
**Official Blind Review #3**

**Rating:** 8

**Review:**

The paper addresses the task of learning temporally stable features for point clouds with an application to upsampling point clouds. Learning point-based descriptors has been a major topic of research in the recent vision and graphics meetings, where approaches have been proposed focusing semantic labeling, geometry-oriented tasks (e.g. normal estimation), and point-based graphics. However, as the paper states, and to the best of my knowledge, no methods have been proposed to learn features in fourth dimension in a temporally stable way. Thus, the very topic of research is significantly novel and promising.

The authors consider a combination of loss functions and train a neural point-based network to learn the features. To stabilize training, the authors carefully study the effect of a series of loss functions, including well-known EMD, losses ensuring slow changes in positions, velocities, and accelerations, as well as a mingling loss to ensure a more uniform spatial point distribution on the output shape. The studied losses are very logical to implement as one aims to ensure that the output should satisfy a temporally smooth motion pattern.

The experimental results provide a clear view of the proposed approach and demonstrate that combining the studied objectives function with a known point-based learning approach leads to a temporally stable feature representation per-point. An ablation study further helps to validate the proposed approaches step-by-step.

To sum up, I believe paper should clearly be accepted, as (1) the work addresses a novel point-based learning task, (2) the research methodology is convincingly presented, and (3) the results provide a clear demonstration of the feasibility of the proposed task.


**Experience Assessment:**

I have published one or two papers in this area.

**Review Assessment: Checking Correctness Of Derivations And Theory:**

I assessed the sensibility of the derivations and theory.

**Review Assessment: Checking Correctness Of Experiments:**

I assessed the sensibility of the experiments.

**Review Assessment: Thoroughness In Paper Reading:**

I made a quick assessment of this paper.

---

> ### Author Response · Authors · 2019-11-13
> **Thanks for your review.**
>
> We thank you for the positive comments and assessment of our work.

---

### Official Review · AnonReviewer1 · 2019-10-30
**Official Blind Review #1**

**Rating:** 6

**Review:**

The paper tries to learn temporally stable representations for point-based data sets and focus on varying size and dynamic point sets, and demonstrate its usefulness in the context of super-resolution. To deal with a difficult target that dynamically moves and deforms over time with variable input and output size, they take a novel temporal loss function for temporally coherent point set generation and siamese network setup for temporal loss calculation. Their novel temporal loss is based on EMD to minimize differences between an estimated point cloud and a desired super-resolution point cloud. The discussion and evolution on multiple loss functions are mostly well done. Except spatial loss is considered, taking the ground truth acceleration and estimated velocity into account is beneficial to this task. Their main contribution is taking permutation invariant loss terms and a siamese training setup and generator architecture, enabling improved output variance by allowing for dynamic adjustments of the output size, and identifying a specialized form of mode collapse for temporal point networks.
They perform an empirical study of their temporal loss function on the generated data set and apply the proposed method to some complex 3D models to conclude the superior performance of temporal loss formulation in contrast to previous work.

Overall, this paper has some significant points on point cloud super-resolution, with the caveat for some clarifications on the theory and experiments. Given these clarifications in an author response, I would be willing to increase the score.

When the input moves slowly enough, the point cloud can be considered static. Can the proposed temporal loss outperform other works under this condition?

Only one previous work PU-Net based on PointNet++ is compared in the paper, I would like to see more discussion on applying the proposed temporal loss with other point-based algorithms.

I am very curious about the effect on different choices of weighting terms hyperparameters in temporal loss and predefined upsampling factor r.



**Experience Assessment:**

I have read many papers in this area.

**Review Assessment: Checking Correctness Of Derivations And Theory:**

I assessed the sensibility of the derivations and theory.

**Review Assessment: Checking Correctness Of Experiments:**

I did not assess the experiments.

**Review Assessment: Thoroughness In Paper Reading:**

I read the paper thoroughly.

---

> ### Author Response · Authors · 2019-11-13
> **Thanks for your review.**
>
> We thank you for the positive review and feedback. Below are our responses to the concerns mentioned in your review.
>
> Slow input movement:
> - When generating point clouds without temporal constraints, one of the biggest problems is a high frequency jittering that occurs due to accumulated small scale inference errors. Especially with very small movements these jittering artifacts are very noticeable. Thus a temporal constraint is also relevant for an almost static input.
>
> Other point-based algorithms:
> - That's a very interesting thought. In theory, our method can be used with other point-based methods. Only the training process would need to be updated, while the method for inference could stay the same. We will add this as a discussion to our document.
>
> Different choices of weighting terms:
> - A larger temporal loss factor leads to central clusters of points. The effects of the different ratios within the temporal loss (velocity and acceleration) are shown in the provided video starting from 0:26. There one can clearly see how the additional acceleration loss term prevents flickering.
>
> With an increased upsampling factor the memory utilization and the training time increases considerably, especially for the volumetric data. As we primarily target temporal stability, we did not evaluate larger upsampling factors so far (smaller ones would be less problematic).

---

### Decision · Program_Chairs · 2019-12-19

**Decision:**

Accept (Spotlight)

**Comment:**

This paper provides an improved method for deep learning on point clouds.  Reviewers are unanimous that this paper is acceptable, and the AC concurs.